# Revealed Preferences of Fourth Graders When Requesting Face-to-Face Help While Doing Math Exercises Online

**Roberto Araya \*** [ID] **and Raúl Gormaz**

Centro de Investigación Avanzada en Educación, Instituto de Educación, Universidad de Chile, Santiago 8320000, Chile; raulgormaz@gmail.com
\* Correspondence: roberto.araya.schulz@gmail.com

**Abstract:** When in doubt, asking a peer can be very helpful. Students learn a lot of social strategies from peers. However, stated preference studies have found that for elementary school students with math questions, they prefer to ask their teacher for help. In this paper, we study revealed preferences instead of stated preferences. We analyzed the behavior of fourth-grade students seeking face-to-face assistance while working on an online math platform. Students started by working independently on the platform, before the teacher selected two or three tutors from among those who have answered 10 questions correctly. Each student was then able to choose between the teacher or one of these tutors when requesting assistance. We studied the students' preferences over 3 years, involving 88 fourth-grade classes, 2700 students, 1209 sessions with classmate tutors, and a total of 16,485 requests for help when there was an option to choose between a teacher or a classmate. We found that students prefer asking classmates for help three times more than asking their teachers when given the choice. Furthermore, this gap increases from the first to the second semester. We also found that students prefer to request help from classmates of the same sex and of higher academic performance. In this context, students from the two highest tertiles sought help from classmates in the same two tertiles, and students from the medium tertile prefer to seek help from students of the highest tertile. However, students in the two lowest tertiles do not prefer asking for help from students from the top tertile more than from their own tertiles.

**Keywords:** help seeking; same-age peer tutoring; revealed preferences; data analytics

## 1. Introduction

Individuals imitate others who are successful [1,2]. With the ecological conditions typical of our hunter-gatherer ancestors, children learned by mimicking peers and adults. It was very rare that a child would learn something on their own, completely independent from the rest. It is very risky to learn what to eat on your own and almost impossible to discover fire and cooking by yourself. Nowadays, most of what preschool children learn is through mimicking. Similarly, teenagers learn about accents, slang, fashion, and dancing by imitating their peers. For example, children of immigrants learn the language of their home country with ease and speak with the accent of their peers rather than their parents [2]. This saves both time and lives. Humans have many unique cognitive abilities. However, these are mostly thanks to having the social cognitive skills needed for participating and exchanging knowledge with others [3]. For example, there is ample empirical evidence suggesting that children prefer helping and learning from individuals based on cues of competence, skill, success, prestige, and similarities with the learner, such as sex [4]. Moreover, experimental work suggests that people give more importance to knowledge acquired from others than from their own observations. This is especially true as situations become more uncertain and problems become more difficult. By imitating and also recombining imitated strategies, we generate and accumulate knowledge that gives us enormous advantages. These advantages more than compensate for the huge

additional energy expenditure that is required by having a brain big enough to handle such complex tasks.

However, due to the enormous advances in cultural evolution over the last 5000 years, today we need students to learn completely new things. These are often of a totally different nature and some of them are very counter-intuitive. We are therefore faced with an evolutionary mismatch, as the human brain has evolved to learn the knowledge and skills required for the life of hunter-gatherers rather than how to read, write, and do math [5,6]. These new skills require an enormous rewiring of several brain areas [7,8]. Thus, in addition to mimicking and recombining, we need other tools that can help us learn an overcrowded curriculum, and do it in just a couple of decades. Enhanced collaboration strategies are critical for overcoming this challenge. Even though other primates are strongly motivated to help others [9], there are significant differences in how help is given. For example, from a young age, children help others with what they need, while chimpanzees help others with what they want. When children know that what the other is requesting would not actually help, then they give what is needed instead of what is requested [9].

Through language, humans have another powerful learning and collaboration strategy: asking questions. In a sample of children aged 2 to 5 years, Chouinard [10] estimated that they ask approximately one question per minute at home. This is an attitude that marks a key difference between human beings and all other primates [11]. A critical decision for students, therefore, is who they should ask. One possibility is the teacher, who knows more and knows how to teach. Another possibility is a classmate. Seemingly, a rational student should always choose the teacher. However, it may be that students feel more comfortable asking their peers for help than asking their teacher. Nevertheless, they may feel too intimidated to ask the best students in the class. A friend who knows just a little more about the subject may therefore be the best option when it comes to asking for help.

Students may have many questions in the classroom. However, very few of these are actually asked. In a study of 710 videos of elementary and junior high math classes, the average number of questions asked by students in a session is less than one [12]. Could it be that students have no more questions? Or, is it that the structure of the school, and in particular, of the classroom, inhibits seeking help and asking questions? Given the typical individual assessment structure, one could argue that classrooms promote competition between individuals. Individual assessments are translated into grades, and grades, whether public or not, make comparisons inevitable. This structure produces a social dynamic that may inhibit collaboration. As a result, this structure may be hindering social learning. One of the advantages of online platforms is that each student can ask for help without exposing themselves to the whole class. But in that case, who do they ask for help? In this paper, we study the data log of ConectaIdeas [13–15], an online platform where students can click a button to ask for help and then choose between the teacher and a list of classmates.

*Related Work*

There are several studies of help seeking and peer tutoring. Lavasani et al. [16] studied help-seeking behavior among 40 female freshmen students at two high schools. The results from pre- and post-tests and student questionnaires revealed that collaborative strategies decreased mathematics anxiety and increased help-seeking behavior. Moliner et al. [17] studied the effect of peer tutoring on mathematics self-concept among 376 students from grades 7 to 9 (12 to 15 years old). They found that same-age and reciprocal peer tutoring may be very beneficial for middle school students' mathematics self-concept.

Marais et al. [18] conducted a single case study, using semi-structured interviews, questionnaires, and observations before, during, and after two mathematic lessons with 20 eighth-graders. They found that the teacher had detailed knowledge of her learners' intended help-seeking behavior. Thurston et al. [19] studied reciprocal peer tutoring in mathematics among 487 10- to 12-year-old students from 20 elementary schools. They

found that the greatest attainment gains were predicted by students having a higher opinion of the cognitive ability of their tutoring partner, as well as having a tutoring partner they believed to be less popular. In another experimental study with 295 11- to 13-year-old students, from 12 classrooms, Thurston et al. [20] found that gains were significantly greater on independent standardized reading comprehension tests for those engaged in peer tutoring. They also found that the gains were greater for the tutors than the tutees. Studying a U.S. sample of 15-year-old students from PISA 2012 (the most recent PISA assessment in which the main area of focus was mathematical literacy), Osborne [21] found that student help-seeking behavior has a statistically significant positive effect on all measures of student achievement in mathematics. Not seeking help when needed put students at a disadvantage, and represents an intention to cease or avoid engagement [22]. Russo et al. [23] found that grade 5 and 6 students valued the opportunity to work with peers and that the students who participated in the focus group interviews expressed that peer explanations were often clearer and more comprehendible than teacher explanations. Newman [24] administered questionnaires to 177 3rd, 5th, and 7th graders about attitudes towards help seeking. Their results revealed that at Grades 3 and 5, the child's expressed likelihood of seeking help was explained by intrinsic preference for challenge, extrinsic dependence on the teacher, and attitudes towards the benefits of help seeking. In a related study, Newman et al. [25] found that students in math classes prefer seeking help from teachers. In another study, Newman [26] analyzed help-seeking behavior among 113 third and sixth graders. They found that sixth graders were more likely to ask the teacher process-related questions, rather than just asking for the answer. On the other hand, in an intensive observational study in mathematics classes with 10 1st, 10 3rd, and 10 5th grade students, Nelson-Le Gall et al. [27] did find that children request more help to peers than to their teachers or to impersonal sources, and did not find sex differences in the number of questions asked per minute.

In a meta-analysis of findings from 50 independent studies of peer tutoring programs in Mathematics at multiple educational stages, Alegre-Ansuategui [28] found that 88% of these programs have positive effects on the participants' academic performance.

A common method for studying help seeking is through surveys. However, there are potential difficulties when using questionnaires. For example, given the low reliability of indices of help-seeking behavior based on surveys, the OECD [29] (p. 368) decided not to include these types of indices in their international database. The internal consistencies fell short of the standards set for indices used in PISA.

There are also other studies that use log data from online platforms. Ogan et al. [30] studied collaboration among students in several Latin American middle schools using a Mathematics Cognitive Tutor (CT) developed at Carnegie Mellon University. Students can ask the platform for help, but cannot ask for help from their classmates or the teacher. However, classroom observations revealed that suburban American middle school students using this same tutor system spent 4% of their time talking to the teacher or another student about the task. Furthermore, students in the Latin American sample spent much more time seeking help from their classmates. The authors found that that even though the Cognitive Tutor was intended for individual use, it became the object of group activity. Subsequently, the primary source of help was not the platform itself but other classmates. Thus, the authors suggested that learning systems may encourage students to seek help from a classmate who has already mastered the relevant skill. Ogan et al. [31] also found that greater levels of collaboration observed in Costa Rican classrooms may suggest that much help seeking occurs outside of the technology. iHelp is another platform that matches students based on their expertise and preferences [32]. However, this system does not take into account real-time progress in problem solving.

In all of these cases, the platforms do not register who the student is asking for help from. To the best of our knowledge, this paper is the first to analyze real-time data of help-seeking behavior, with student preferences recorded via an online platform.

Our research question therefore asks "Who do students turn to for help in a math class?" We also wonder whether there are any factors that may influence this decision, including whether the students' preferences are influenced by gender and performance.

## 2. Materials and Methods

We used the log data from the collaboration features of ConectaIdeas [13,15]. This is a web-based platform that we started to develop in 2002 with partial support from CORFO, a government agency in Chile. Eight years of data on national standardized test scores for every school in a low SES district has shown positive effects [14,33,34]. Furthermore, two independent Randomized Controlled Trials (RCT) have also found positive learning effects. One RCT was a year-long treatment in 48 classes and using the National Standardized test [35] as a post-test. The other RCT was a half-year treatment. It used third-party standardized tests and was implemented during a period of significant social turmoil [36]. Several other interventions to enrich the math and STEM lessons have been implemented with ConectaIdeas [37,38] with positive results.

In ConectaIdeas, the teacher selects from a list of multiple choice or other closed form exercises, as well as open-ended questions. A mix of these questions is then assigned for the session. Teachers access the system through a tablet or smartphone in order to track the students' participation and performance during the session (Figure 1). ConectaIdeas also helps the teacher analyze responses to multiple-choice and open-ended questions in real time.

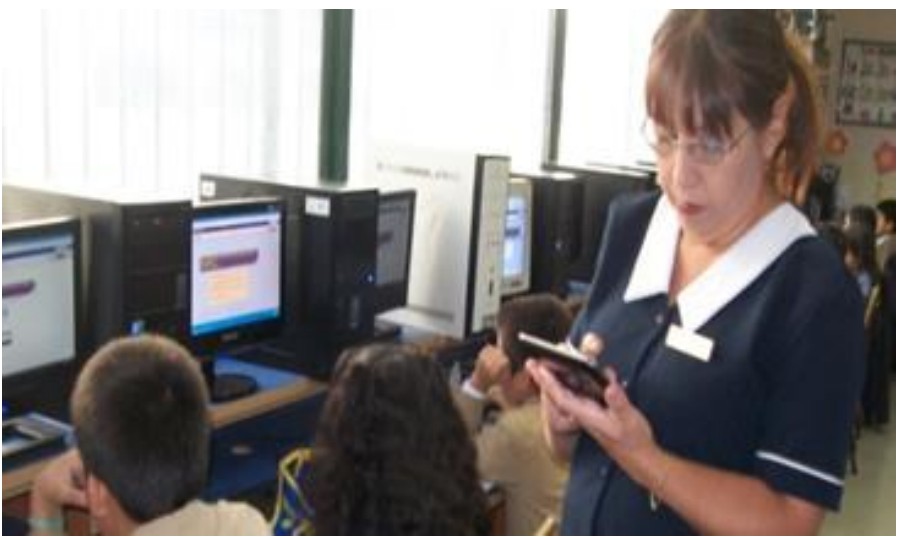

**Figure 1.** Teacher reading the ConectaIdeas Early Warning System on a tablet.

An early warning system constantly lists students who are having more difficulties (Figure 2). The teacher therefore knows which students need personal attention and on which specific exercises. All of this information is recorded for future analysis. The early warning system also detects whether there are any exercises causing problems for the whole class. This way, the teacher or the lab coordinator can freeze the system and then explain the required concepts. ConectaIdeas is designed to drive whole-class progress and not to leave students to their own devices.

## Casos ordenados por prioridad

# Prioridad   ⏱Demora (min)

| | | | Ejercicios | | |
|---|---|---|---|---|---|
| # | Profesor | Estudiante | omitidos | realizados | Desempeño en realizados |
| ▶ | | 🟢 Student2 | 12 | 9 | 4.1 |
| | | 🔴 Student3 | 0 | 0 | 0.0 |
| ▶ | | 🔴 Student9 | 0 | 5 | 0.0 |
| ▶ | | 🟢 Student7 | 1 | 6 | 5.8 |
| ▶ | | 🟢 Student8 | 0 | 7 | 7.0 |
| ▶ | | 🟢 Student1 | 0 | 20 | 7.0 |
| ▶ | | 🟢 Student4 | 2 | 31 | 6.2 |

**Figure 2.** Teacher dashboard ranking students based on a real-time estimate of how much help they need. This is displayed via the ConectaIdeas Early Warning System. It allows the teacher to monitor the whole class' progress during the session and to assign a teacher or a student to help.

ConectaIdeas is designed to promote cooperation between students. In this context, it has several tools that aid collaboration. Firstly, the teacher can review the answers in real time and ask students to peer review the answers to open-ended questions. Secondly, it has the option to organize tournaments between classrooms, whether from the same school or across different schools. These are mathematics competitions where each class gains a score, similar to a soccer or basketball match [39]. It is designed to increase collaboration between classmates. This way, the whole class works on preparing for the tournament in order to out-perform the other classes. Thirdly, ConectaIdeas has a peer tutoring option. In each class, students who correctly complete 10 exercises are automatically assigned to a list of potential tutors for the session (Figure 3). From this list, the teacher then selects two or three students to act as tutors for the rest of the session (Figure 3).

## Alumnos que pueden ser monitores a las 09:20:36

**Criterios** Cantidad mínima: 10, Errores: 0 y Aciertos: 100%.

| Estudiante | Convertir en monitor |
|---|---|
| Student1 | 👤⭐ |
| Student3 | 👤☆ |
| Student4 | 👤☆ |
| Student5 | 👤⭐ |
| Student8 | 👤☆ |

**Figure 3.** List of students preselected by ConectaIdeas as potential tutors. The students highlighted in orange have been selected as tutors by the teacher.

From this point on, students can ask for help from the teacher or a peer tutor by selecting their name from a menu (Figure 4). Before this point, students are only able to

ask for help from the teacher. Once help has been provided, the tutees are then given the option to rate the quality of the support they received. Using this information, the teacher can work with the tutors to improve the understanding of the content and their communication skills.

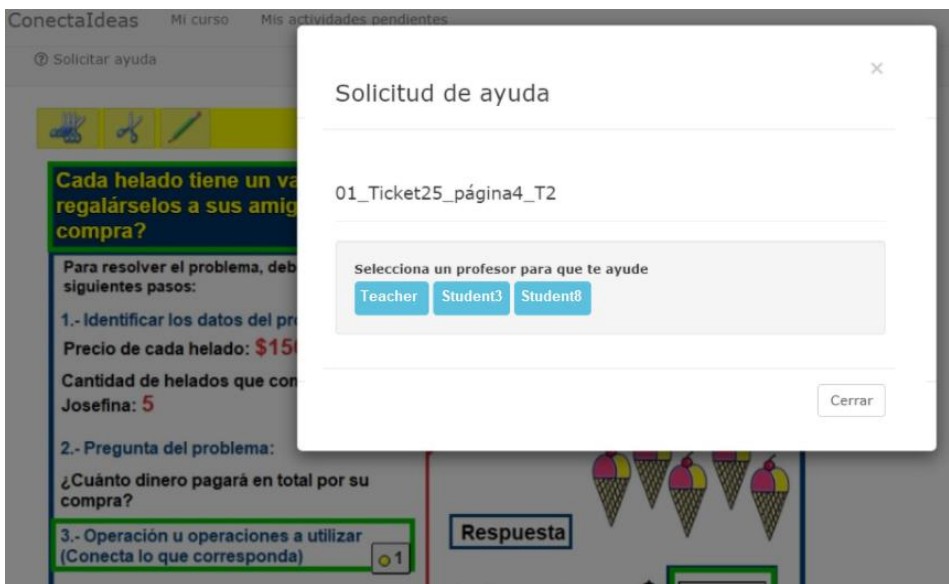

**Figure 4.** Pop-up window where the student asks for help and chooses who to ask.

In this study, we analyzed the data logs from this help-seeking component of ConectaIdeas. All schools are urban schools and the students are of a Low Socioeconomic Status (SES). In several countries, there is much interest in improving student academic performances in Low SES urban schools. The data collected here are part of a project to improve performance in these types of students.

One important methodological factor is the framing effect. According to Goldin et al. [40], people's choices depend on seemingly arbitrary features of the decision-making environment. This includes factors such as which option is the default, the order in which options are presented, or which features of the decision are most salient. Such framing effects cast doubt on the traditional approach to revealed preference, which equates choices with preferences. In our case, students seeking help have to select from a list of three or four people, with the teacher always on the far left of the list. This order can affect the selection, generating a bias in favor of the teacher when requesting help.

For each session, we calculated revealed preferences in the following way. Given two groups $G_i$ and $G_j$ (which can be teachers and students, male students and female students, or students at different performance levels), we counted the size of each group, $Card(G_i)$; measured the length of time T during the session where students had the option to choose between tutors from group $G_i$ and group $G_j$ when requesting help; counted the number $N_{ij}$ of help-seeking events among students from group $G_i$ to tutors of group $G_j$; and then calculated the preference $P_{ij}$ that members of group $G_i$ chose a tutor from group $G_j$ as:

$$P_{ij} = \frac{N_{ij}}{T\, Card(G_i)}. \tag{1}$$

Thus, $P_{ij}$ as the number of times per student from group $G_i$ per hour that a tutor from group $G_j$ was selected when there was the option to select a tutor from $G_i$ or $G_j$.

For example, consider $G_i$ to be the male students in a classroom and $G_j$ to be the female students in the same classroom. In this case, $P_{ij} = 2$ means that, on average, a male student asks for help from female tutors two times per hour, assuming there is at least one male and one female tutor available on the ConectaIdeas platform.

## 3. Results

In this study, we gathered help request data from 2700 fourth grade students (1301 girls, 1399 boys) from 88 classes and 33 schools. All of this data was recorded on the ConectaIdeas platform. The mean age was 9.8 years old with SD of 0.74 years.

It is important to highlight that when comparing requests to teachers with requests to peers, we only counted help-seeking events when there was an option to choose between the teacher and a classmate, since there were more requests for help to the teacher in other moments of the session or in other sessions. However, during those moments there was no choice. On the platform, there were a total of 56,246 help requests, but only 16,485 requests when there was a choice between requesting a teacher and at least one student. Of those requests, 4112 were made to teachers and 12,373 were made to classmates. Each student requested help from another student (SS) 0.475 times per hour, but only requested help from the teacher 0.151 times (ST) (Figure 5). This means that when peer tutors are available, students prefer asking classmates for help 3 times more than asking their teachers. More precisely, the ratio of preference to classmate to teacher is 3.2. However, this ratio is 2.5 in the first semester and 3.6 in the second semester. If we compute for each session the ratio of the number of help requests to students over the number of help requests to teachers, then the average of these ratios for the first semester is 3.73 and for the second semester it is 4.84. The difference is statistically significant with a $p$-value of 0.0012. This means that, that as time passes and students get used to the ConectaIdeas platform and asking classmates, their preference for asking their classmates for help increases.

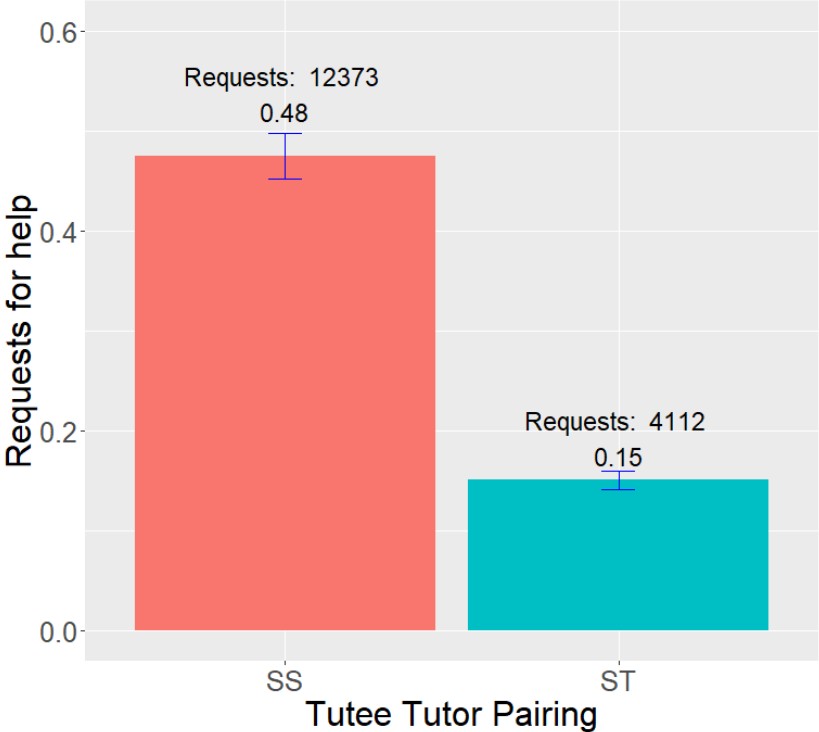

**Figure 5.** Number of help requests per student per hour when there is an option to choose between a teacher or a classmate. There were 12,373 requests for help to students (SS) and only 4112 requests for help to teachers (ST). Each hour when there was choice, each student requested help from a classmate 0.475 times, whereas they only requested help from a teacher 0.151 times. The difference is statistically significant, as can be seen from the Confidence Intervals.

We found that the help seeking events to another student increases as the number of student tutors in that session increases. Similarly, help seeking events to a teacher increases as the number of teacher increases. Table 1 shows, in different configurations of the number of student tutors and teachers, the mean of help requests to students (SS) per hour, and the mean of help requests to teachers (ST) per hour.

**Table 1.** Number of help requests per student per hour for different configurations of the number of teacher and number of student tutors.

| SS | 1 Student | 2 Students | 3 Students |
|---|---|---|---|
| 1 Teacher | 0.26 | 0.51 | 0.43 |
| 2 Teachers | 0.27 | 0.60 | 0.67 |
| 3 Teachers | 0.36 | 0.51 | 0.66 |
| **ST** | **1 Student** | **2 Students** | **3 Students** |
| 1 Teacher | 0.11 | 0.10 | 0.10 |
| 2 Teachers | 0.16 | 0.14 | 0.18 |
| 3 Teachers | 0.26 | 0.16 | 0.26 |

From Table 1, we can also see that for each configuration of students and teachers available, the help requests per unit of time to students (SS) are greater than to teachers. Of all the situations where there was a choice, the most frequent was two teachers and two student tutors. In that case, the number of help requests to students per hour was 0.60 and the number of help requests to teachers was only 0.14. The second configuration with high frequency was two teachers and one student tutor. In that case the number of help requests to students per hour was also higher than the number help requests to teachers. Even in the extreme case of having three teachers available from whom students can request help and only one student tutor available, the number of requests to students per hour (0.36) was also higher than to teachers (0.26).

On the other hand, there were 6844 help-seeking events when a choice could be made between male and female peer tutors. There were 3261 help-seeking requests made to female classmates and 3583 made to male classmates. The data reveals that girls prefer asking girls for help, while boys prefer asking boys (Figure 6). The girl to girl help requests rate was 0.409 per hour when there was an option to choose between a female or male tutor. We had 2273 such events. The boy to boy help requests rate was 0.392 per hour when there was an option to choose between a female or male tutor. We had 2451 such events. On the other hand, the girl to boy help requests rate was 0.207 per hour and boy to girl help requests rate was 0.182 per hour, when there was an option to choose between a girl and a boy, and the number of such events were 1132 and 988, respectively. In addition, on average girls sought help more frequently than boys, but this difference was not statistically significant.

To study the effect of students' performance, we classified the historical performance of the students on the ConectaIdeas platform into three tertiles. The first tertile (L) represents the lowest performing students in the class. The second tertile represents the medium performing students in the class (M). The third tertile represents the highest performing students in the class (H). Table 2 shows the number of times tutors from the different tertiles were selected. In total, 12,481 requests were made when tutors from different tertiles were available. Notably, 941 requests were made to students from the first tertile, 2870 to students from the second tertile, and 8670 to students from the third tertile. Thus, higher tertile students received more requests for help. But this may be because the teacher selects students from the highest tertile as tutors many more times, and selects students from the lowest performing tertile much less often. On the other hand, the students who most requested help are in the lowest tertiles.

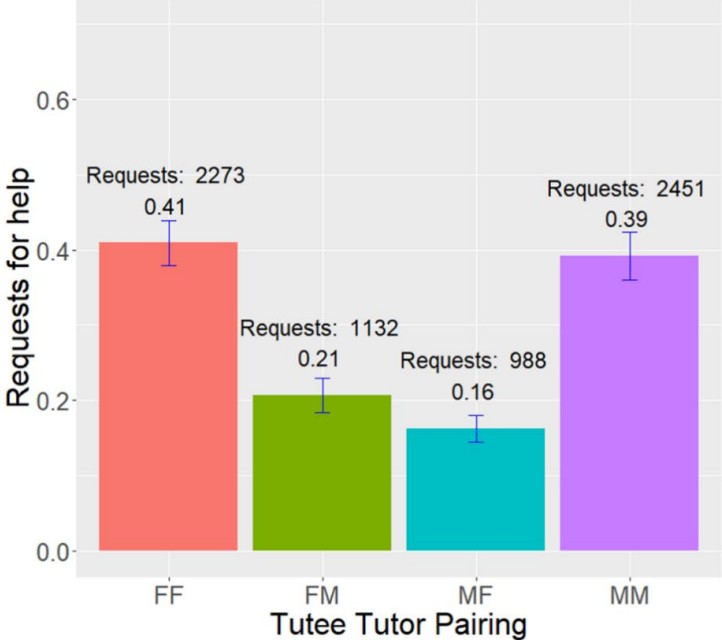

**Figure 6.** Number of help requests per student per hour when there is an option to choose between a male and female classmate, broken down based on the sex of the tutor and tutee.

**Table 2.** Total number of requests for help from student to student tutors of the indicated tertiles.

|  | **TUTOR L** | **TUTOR M** | **TUTOR H** | **Total** |
|---|---|---|---|---|
| STUDENT L | 393 | 1138 | 3273 | 4804 |
| STUDENT M | 294 | 934 | 2904 | 4132 |
| STUDENT H | 254 | 798 | 2493 | 3545 |
| Total | 941 | 2870 | 8670 |  |

On the other hand, note that the numbers of help requests per session per student have a very regular pattern (Table 3). When moving one square to the right, a factor of approximately 3 is applied. Also, when moving up one square, a factor of approximately 1.2 is applied.

**Table 3.** Number of help requests per session per student of the indicated performance to a tutor of the indicated performance, per session.

|  | **TUTOR L** | **TUTOR M** | **TUTOR H** |
|---|---|---|---|
| STUDENT L | 0.018 | 0.056 | 0.168 |
| STUDENT M | 0.015 | 0.043 | 0.146 |
| STUDENT H | 0.011 | 0.034 | 0.112 |

From Figure 7, we can see that when there is an option to seek help from students of the lowest tertile or the two highest tertiles, students from the two highest performing tertiles, MH, prefer asking for help from classmates in the same two tertiles (Figure 7). They ask 0.34 requests per hour instead of 0.21 requests per hour to students belonging to the lowest tertile. The difference is statistically significant.

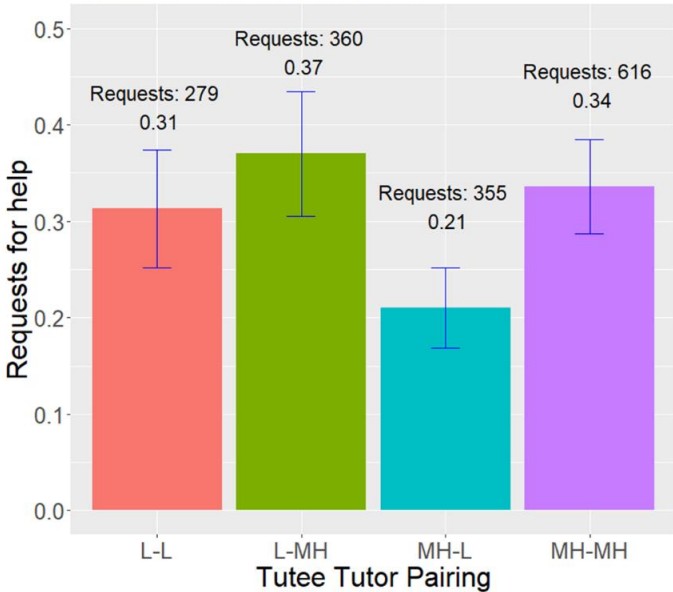

**Figure 7.** Number of help requests per student per hour when there is an option to choose between students of different performance levels, broken down based on performance level by tertile.

Similarly, students from the two lowest performing tertiles, LM, prefer asking for help from classmates in the same two tertiles (Figure 8), but this difference is very small and it is not statistically significant.

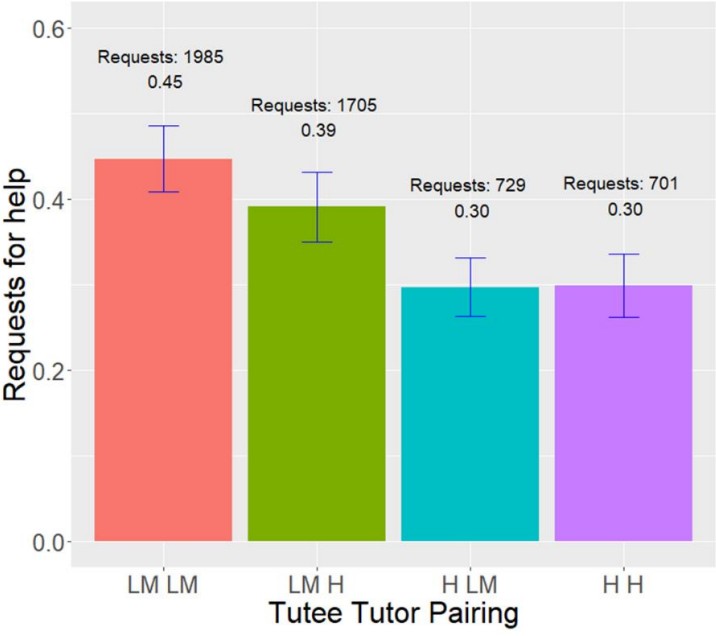

**Figure 8.** Number of help requests per student per hour when there is an option to choose between students of different performance levels, broken down based on performance level by tertile.

Additionally, from Figure 9, we can see that students of the highest tertile prefer to seek help from students from their tertile. However, even though students from the lowest tertile (L) do seem to prefer seeking help from tutor students from the highest tertile (H) over the second highest tertile (M), this is a slight difference and it is not statistically significant.

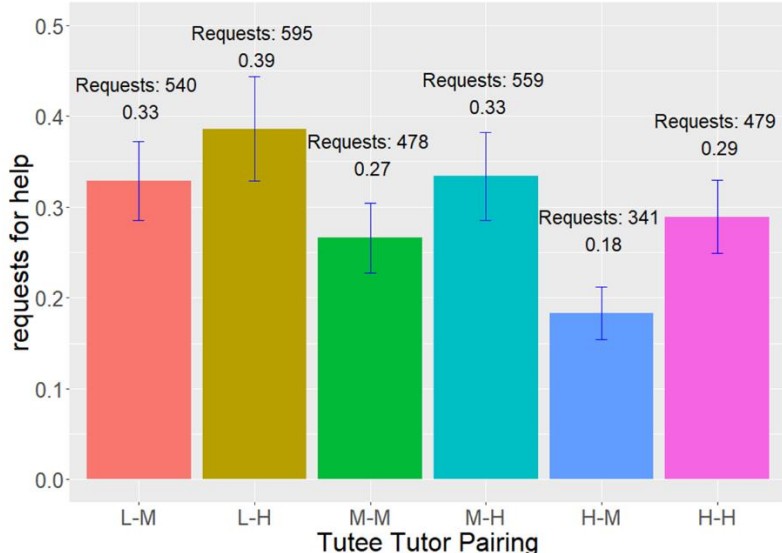

**Figure 9.** Number of help requests per student per hour when there is an option to choose between students of different performance levels, broken down based on performance level of the two highest tertiles.

The above analysis is by tertiles of performance. It seems to indicate that in general, students prefer to seek help from better performing tutors. To confirm this conjecture, we considered the sessions in which there were only two tutors. There were 493 such sessions. In those sessions, we counted the requests for help from students whose performance is between the performances of the two tutors. There were 242 sessions where such events occurred. In them, there were 588 requests for help from a student to a tutor with a lower performance than the tutee, and there were 761 requests for help to tutors with a higher performance than the tutee. That is, for students with a performance between the two tutors, the percentage of requests to tutors with the lower performance than the tutee was 43.69%, and the percentage of request to tutors with a higher performance than the tutee was 56.41%. This difference is statistically significant.

On the other hand, when the tutee's performance is lower than that of the two tutors, the tutee prefers the tutor with the highest performance 51.41% of the time, but this percentage is not statistically significantly higher than the percentage of times the tutee prefers the tutor with the lowest performance. In contrast, when the performance of the tutee is higher than that of the two tutors, the tutee prefers the tutor with the highest performance 54.47% of the time. This percentage is statistically significantly higher than the percentage of times that the tutee prefers the tutor with the lowest performance.

Accounting for the opportunities to ask students of different tertiles for help, students prefer to seek help from better performing tutors. For example, students in the two higher tertiles prefer to ask a student of their tertile. However, students in the two lowest tertiles were 15% more likely to ask a student for help from their tertile than a student in the highest tertile, although in this case the difference is not statistically significant.

## 4. Discussion

We studied the help-seeking behavior among 88 fourth grade classes from 33 schools, analyzing a total of 1209 sessions with tutors out of 3258 sessions. There were 16,551 help-seeking events when there was a choice between a teacher and tutors. A total of 12,436 were help requests to classmates and only 4115 to teachers. For our analysis, when we compared requests for help to teachers or student tutors, we only considered help-seeking events when students could choose between requesting help from their teacher or from another student. By doing so, we were able to access the revealed preferences of fourth-grade students when choosing between the two for requesting face-to-face assistance in an online session doing math exercises. Within the first 10 min of each session, the teacher selected two or three tutors from a list of students who have answered 10 questions correctly. Students could then choose between the teacher and one of the listed tutors when requesting assistance. When in doubt, asking a peer can be very helpful. Students learn a lot of social strategies from peers. However, for students with math questions, it could be more effective to ask the teacher. Indeed, stated preference studies have found students prefer to ask for help from their teacher. However, when considering the time periods when students could choose who to ask for help, we found that students prefer asking classmates for help three times more than asking their teachers. Furthermore, this gap increases from the first to second semester. We also found that students prefer to receive help from classmates of the same sex and of higher academic performance. In this context, students from the two highest tertiles sought help from classmates in the same two tertiles, and students from the medium tertile prefer to seek help from students of the highest tertile. Moreover, tutees with performance between the two tutors prefer to seek help from the tutor with a higher performance. However, students in the two lowest tertiles preferred asking for help to students from their tertiles rather than the top tertile, but the difference was not statistically significant.

The teacher appears further to the left in terms of the buttons to select who to ask for help from and since in Spanish this is read from left to right, the teacher is always read first. This can generate a framing effect that causes the teacher to be selected rather than a classmate. However, despite this potential bias, we found that, at times when there is the option to choose between the teacher and students when requesting help, students choose to request help from a classmate three times more often than from the teacher.

Our findings on revealed preferences are completely opposite to the findings of stated preferences [25], who found that students in grade 3, 5, and 7 math classes prefer seeking help from teachers. A possible explanation of this difference could be due to the nature of asking questions in an online platform. In typical offline classroom environments, Newman et al. [25] found that students see the teacher less likely to think they are "dumb" for seeking help. In the ConectaIdeas platform, seeking help mechanics are a more structured part of the platform, and thus it could be more socially acceptable to ask a peer. Another possible explanation for this difference could be due to the fact that stated preferences are based on surveys and rely on respondents making choices over hypothetical scenarios, whereas revealed preferences are based on observations on actual choices made by people. In [25], students were interviewed individually using a structured questionnaire. This second explanation agrees with the intensive observational study in mathematics classes of 10 3rd, 10 4th and 10 5th grade students [10], where they found that children request more help from peers than from their teachers.

For future work, we plan to study the satisfaction of tutees and tutors. Since the ConectaIdeas platform has the facility that each tutee can optionally rate the help received, and each tutor can also optionally rate the degree to which he estimates that the tutee has understood them, then there exists in the platform all the information to perform such a study. Another plan for the future is to analyze the tutees' learning as a result of the aid received. Here it is interesting to study if there are differences when the tutor is the teacher or a classmate, and also analyze how learning depends on the historic math performance of the tutor and the performance gap between the tutor and tutee. We also plan to deepen

the study of requests for help from students in the lower terciles. We believe that it is important to better understand the preferences of these students and what motivations may be influencing their preferences. Finally, it would be important to consider the framing effect. If we could manipulate the order when teachers and tutors appear on the screen, we could easily estimate the effect of the order on the revealed preferences.

## 5. Conclusions

Thanks to our capacity for complex collaboration, language, and active questioning, humans have been able to accumulate and increase their knowledge. Much of what we learn is through interactions with others. Children can acquire cultural traits not only from their parents (vertical transmission) but also from nonparental adults (oblique) and peers (horizontal) [41]. However, to enhance social learning, it is critical to ask questions. But in the classroom, it is mainly the teacher who asks questions. There is already a long history of studies of teacher questions. In 1912, Stevens [42] considered that "the subject of questioning should have a place in the training of every teacher—a place that is comparable in importance with 'fund of knowledge' and psychology" Stevens [42] (p. 5). categorized teachers' questions, producing statistics in six different subjects, including middle and high school science and mathematics. In the aforementioned study, the number of questions used per session was counted, as well as the number of "memory" questions. The study also considered the quality of the questions. But to enhance social learning, it is critical that students also ask questions, and in particular to ask classmates for help. This is a truly horizontal form of learning.

We need to promote collaboration and have students ask each other for help. With technology we can ensure that tutors know the content reasonably well and, on the other hand, that each student can privately select whom to ask for help. Moreover, with data analytics we can automatically count the help seeking behavior, complementing the study of teacher questions that started a century ago. In this scenario, and contrary to the studies of stated preferences, the revealed preferences show a systematic behavior to ask classmates. Despite the fact that the teacher knows more, the students prefer to ask questions to classmates, and of the same sex and higher performance. This behavior makes sense within an evolutionary framework. Hunter gatherer early and middle childhood children learn mainly playing and imitating from same age or slightly above the same age children, and in some learning tasks they learn from other children of the same sex [43]. In any case, it would be important to verify these findings in students of other levels, in other subjects, and in other countries.

**Author Contributions:** Conceptualization, R.A.; Data curation, R.G.; Formal analysis, R.A. and R.G.; Funding acquisition, R.A.; Investigation, R.A. and R.G.; Methodology, R.A. and R.G.; Project administration, R.A.; Resources, R.A.; Software, R.G.; Supervision, R.A.; Validation, R.A. and R.G.; Visualization, R.G.; Writing–original draft, R.A.; Writing–review & editing, R.A. and R.G. Both authors have read and agreed to the published version of the manuscript.

**Funding:** This research was founded by the Chilean National Agency for Research and Development (ANID), grant number ANID/PIA/Basal Funds for Centers of Excellence FB0003.

**Institutional Review Board Statement:** Ethical review and approval were waived for this study, due to it being a class session during school time. The activity was revised and authorized by the respective teachers.

**Informed Consent Statement:** Student consent was waived due to authorization from teachers. Given that there are no patients but only students in a normal session in their schools, within school hours, and using a platform that records their responses anonymously, the teachers authorized the use of anonymized information.

**Data Availability Statement:** Due to the nature of our data and the ethical guideline that this project follows, we, unfortunately, cannot share the data set.

**Acknowledgments:** Support from ANID/PIA/Basal Funds for Centers of Excellence FB0003 is gratefully acknowledged.

**Conflicts of Interest:** The authors declare no conflict of interest. The funders had no role in the design of the study; in the collection, analyses, or interpretation of data; in writing of the manuscript, or in the decision to publish the results.

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
