# Peer review of "Revealed Preferences of Fourth Graders When Requesting Face-to-Face Help While Doing Math Exercises Online"

_education, doi:10.3390/educsci11080429_

Round 1

Reviewer 1 Report

Overall  

Thanks for the opportunity to review the paper. Overall, I think it is a valuable contribution, particularly because it affirms the idea that students value learning from each other. This is consistent with research in mathematics education that has found that one of the benefits to teaching mathematics through problem solving is that it introduces a structure whereby students learning through dialogue/ mathematical discussion with peers (for example, see page 221 from the publication: https://www.iejee.com/index.php/IEJEE/article/view/1252.). This can be contrasted with instructional approaches that rely exclusively on explanations from teachers.

The overall headline finding is very valuable (i.e., that students are three times more likely to ask a peer for help than a teacher, as gleaned from revealed preferences), and I will certainly be citing this finding in my own research going forward. However, I am a bit concerned that this gets lost in both the introduction and conclusion. I suggest shortening the former and that you consider deleting the latter. See my further comments below in relation to each of the section.

Introduction

Consider shortening the introduction. Could you begin the paper instead at the line: “Individuals imitate others who are successful.” I think the first section of the introduction is unnecessary, and the key ideas in the paper could be introduced faster.

I am unsure whether many educators in elementary contexts would agree with this statement: “Typical classrooms promote competition between individuals, since assessments are translated into grades, and grades, whether public or not, make comparisons inevitable. This structure produces a true zero-sum game and therefore a social dynamic that may inhibit collaboration. As a result, this structure may be hindering social learning.” Typical elementary classrooms in many countries are not nearly as competitive and grades driven as this statement suggests – and are certainly not best understood as a “true zero-sum game”. Surely completion and collaboration should be viewed as existing in some sort of exchange/ interplay.

Even if one accepts the premise that all students are concerned with are their grades (which might be truer in some countries/ cultures than others), when one considers that it is not only at the individual level that this completion unfolds, the zero-sum assumption amongst peers does not hold. For example, it is also true that a ‘classroom’ at one level might be competing against other classrooms in a school – such that students in a classroom are incentivised to generate positive gestalts to drive all their learning through collaborating in order to enhance their performance vis-à-vis students in other classrooms. Similarly, and more relevantly, a ‘set of classrooms’, that is, a school, competes against other ‘sets of classrooms’. Consequently, student relationships within a classroom should at a minimum be viewed as being governed by a complex set of incentives, even if they are single minded about the outcomes they are aiming for (i.e., grades focussed). (In fact, you even touch on these ideas in your method section when you discuss how the mathematics program has been used).

Results

The last section of the results section looking at student preferences via performance is interesting (i.e., comparing tertiles preferences), but somewhat convoluted to migrate through. It would benefit from a summary at the end of the section. I also feel that this section is perhaps presented in less clear prose than other parts of the paper.

In terms of what to include in the summary, it would be quite useful to be able to make statements such as “accounting for the opportunities to ask students of different tertiles for help, students in the bottom tertile were X more likely to ask a student for help from the middle or upper tertile than a student in their own tertile”. This enables easier comparison with data from the first part of the results section comparing propensity to ask teachers versus students for help.

Having said all this, when the data is summarised in the discussion at the end of page 12/ start of page 13, it is very clear what the key findings were.

Discussion

I do not think the methodology needs to be revisited in so much detail at the beginning of the discussion. Perhaps you could begin with the sentence “We studied help seeking behaviour…”, and deleted the sentences prior to this.

Reference to the SES of the school should be in the method section, unless the authors elaborate on why it is relevant to this particular set of results.

Some statements in the discussion do not seem relevant to the discussion at hand, or at least require more elaboration/ context. For example: “When in doubt, asking a peer can be very helpful. Students learn a lot of social strategies from peers”

Conclusion

In a similar manner to the introduction, I would suggest getting to what is important far earlier. For example, you could delete the first few sentences and begin at: “Thanks to our capacity for complex collaboration, language and active questioning, humans have been able to accumulate and increase their knowledge.”

You should not be introducing new references and new ideas in the conclusions section. I would delete the reference to the 1912 paper and the importance of ‘teacher questioning’. It is only tangentially related to the current findings. Furthermore, the links to an evolutionary framework seem very speculative and unnecessary, particularly given this language and analysis has not been a major part of the paper so far.

Upon reflection, I actually think you should delete the concluding section altogether and finish with the discussion outlining future research directions. This is a stronger note to finish the paper on. My feeling is that the conclusion detracts from what is most interesting and important about the paper.

Language

There are a couple of typographical errors throughout – such as in the Table 1 heading.

Author Response

Thank you very much for your review. Attached is a word file with our answers. 

Reviewer 2 Report

As someone who primarily is focused on peer learning at the tertiary level, I enjoyed reading this remarkable study focused on fourth graders. This was an enormous study with more than thirty schools, so many students, and thousands of behavior choices by the students. You do a good job of explaining the next study to pursue. Since you focused on urban students who were lower SES, I suggest highlighting that more in the manuscript at the beginning. In the U.S., there is much interest in improving student academic performance in urban schools. I suggest you mention that another future study could focus on why the lowest tertile students prefer to interact with each other rather than the highest tertile. I realize that you stated that the difference was not statistically significant. Still, I think it warrants further investigation by someone. 

The only area for improvement is the conclusion. The first half of the conclusion was distracting to me. I was ready to read a focused statement of the findings and suggestions for next steps. The animal analogy at the beginning was not what I was waiting for. The first half of the conclusion could appear in the literature review instead. The final one-third of the conclusion did an excellent job of reminding me as the reader what was the major points that were discovered through the research

Author Response

Thank you very much for your time and dedication to review our manuscript. Attached is a word file with our answers.
